# Impact Hospitality: Creating Social Impact through Hospitality

Clinton W. Mitchell 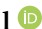

College of Professional Studies, Northeastern University, Boston, MA 02115-9959, USA;
c.mitchell@northeastern.edu

**Abstract:** Hospitality is the relationship between a guest and a host, wherein the host receives the guest with some amount of goodwill, including the reception and entertainment of guests, visitors, or strangers. At the outset of the pandemic, it was clear that the hospitality industry that has been around for centuries would need to be reimagined to survive and perhaps come out stronger. Enter trailblazers Donte Johnson and Jason Bass of Hotel Revival in Baltimore. Their creativity, ingenuity, and compassion sought to effect significant positive change to address the pressing issue of COVID-19 and its devastating effects. The two define hospitality as "just taking care of people," which is what they decided to do when the world closed its doors and they opened theirs. This case study seeks to define a new form of hospitality designed to serve the double bottom line of *profit and people*. This body of work chronicles the stories, lessons learned, and the path ahead for an industry in need of a way forward to create Impact Hospitality.

**Keywords:** impact; hospitality; social impact; COVID-19; coronavirus; sustainability

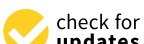



## 1. Introduction

The World Health Organization (WHO) first declared COVID-19 a world health emergency in January 2020. On 11 March 2020, it announced the viral outbreak was officially a pandemic, the highest level of health emergency [1]. Subsequently, it evolved into a global public health and economic crisis that affected the $90 trillion global economy beyond anything experienced in nearly a century [2].

As of this writing, the infection has sickened 487 million people globally and taken the lives of 6.1 million people. During this time, the United States reported over 80 million Americans had been diagnosed and over 978,000 had died from the virus. During the same period, Maryland recorded over 1 million infections and over 14,000 deaths [3]. Over the course of this pandemic, more than 80 countries have closed their borders to arrivals from countries with infections, ordered businesses to close, instructed their populations to self-quarantine, and closed schools to an estimated 1.5 billion children [2].

The impact of the pandemic has been seen and felt across every sector and industry, but perhaps the hardest hit industry has been hospitality. Hotels were one of the first industries affected by the pandemic after travel was ground to a virtual halt in early 2020. The hotel industry experienced the most devastating year on record in 2020, resulting in historically low occupancy, massive job loss, and hotel closures across the country. The impact of COVID-19 on the travel industry so far has been nine times that of 9/11, likely making it one of the last industries to recover [4] (p. 3).

From the outset of this crisis, it was clear that the industry that had been around for centuries would need to be reimagined to survive and emerge stronger. Enter trailblazers Donte Johnson and Jason Bass, of Hotel Revival in Baltimore, Maryland. Their creativity, ingenuity, and compassion sought to effect significant positive change to address the pressing issue of COVID-19 and its devastating fallout. When the world closed its doors, they decided to keep theirs open to support the city and community they vowed to be of, not just in. This case study seeks to introduce a new form of hospitality designed to serve the double bottom line of *profit and people*. This body of work chronicles the stories,

lessons learned, and the path ahead for an industry in need of a way forward to create Impact Hospitality.

## 2. Materials and Methods

Over the course of 6 months, I interviewed Donte and Jason to conceptualize their idea of this new form of hospitality. From there, I gathered baseline data about the city's socioeconomic standing from various sources, including the City of Baltimore, Georgetown University's Beeck Center for Social Impact and Innovation, and proprietary data from the hotel. To assess impact, I interviewed several of the hotel's vendors and suppliers to ascertain how their relationship with the hotel impacted them.

Some of the information and materials utilized for this paper are proprietary property of Hyatt Hotels and are therefore unavailable for release to the public.

## 3. Discussion

*3.1. Hospitality Industry Hit Hard by COVID-19*

"Ultimately, hospitality is just taking care of people."

—**Donte Johnson, GM Hotel Revival**

**Hospitality** hos·pi·tal·i·ty *noun*

The friendly and generous reception and entertainment of guests, visitors, or strangers.

Hospitality, derived from the Latin word "hospes," meaning visitor or stranger, is one of the oldest businesses, going back to the innkeepers and taverns of biblical times [5]. The term means extending a welcome to travelers or offering a home away from home. At its core is the relationship between a guest and a host, wherein the host receives the guest with some amount of goodwill, including the reception and entertainment of guests, visitors, or strangers. Chevalier Louis de Jaucourt described hospitality as the virtue of a great soul that cares for the whole universe through the ties of humanity [6].

While hospitality enjoys a long and storied history, tourism is a more recent invention which began in Europe, with Switzerland being one of the first countries to develop special accommodation and services for travelers. In the late 1800's, the concept of leisure tourism and hospitality spread across Europe, bringing flocks of wealthy travelers to Switzerland.

The hospitality and tourism industry is a vast sector that includes all the economic activities that directly or indirectly contribute to, or depend upon, travel and tourism. This sector includes:

- Hotels & Resorts
- Restaurants & Catering
- Night Clubs & Bars
- Travel & Transportation
- Tourism
- Spas & Wellness
- Cruise Liners & Bus tours
- Events (Private, Business, Cultural, and Sports)

Prior to the pandemic, hotels supported 1 in 25 American jobs—2.3 million direct hotel operations jobs and 8.3 million hotel-supported jobs in total—and contributed $660 billion to the U.S. Gross Domestic Product (GDP) [4] (p. 3). In 2020, more than 670,000 direct hotel industry operations jobs and nearly 4 million jobs in the broader hospitality industry were lost due to the pandemic [4] (p. 4). At the beginning of 2021, at least 2 in 10 hotel employees who were working in March 2020 were still not back on the job at all, while many more were not back working full-time [4] (p. 4).

COVID-19 brought hotel occupancy to a historic low of 24.5% in April 2020. Annual occupancy in the United States fell to roughly 44% for the full year [4] (p. 6). Additionally, the total number of rooms occupied fell by 458 million from 2019 figures [4] (p. 6). This

drastic reduction forced hotels to reduce staff sizes and demanded a consolidation of roles for many team members.

The reduction of guest stays does not only affect the bottom line of the individual property but affects the community at-large as well. Urban hotels are major employers due to their larger-than-average property size, so a reduction in occupancy that spurs a reduction in staff size negatively affects the community. This can be felt financially by state and local governments by way of tax revenues.

In 2019, direct state and local tax revenue, including hotel-specific occupancy taxes, sales taxes, property taxes, and others from hotels in the United States reached nearly $41.1 billion. In 2020, direct state and local tax revenue generated from hotels fell by approximately $13 billion to $27.5 billion in 2020. The numbers are not expected to rebound until at least 2023 [4].

As the world attempts to recover, hotels were expected to add 200,000 direct hotel operations jobs in 2021 but remain nearly 500,000 jobs below the industry's pre-pandemic employment level [4]. With an initially slow vaccine rollout, the advent of new strains of the coronavirus, and the acclimation to remote work, travel is not expected to return to 2019 levels until 2024 [4].

*3.2. Baltimore: Charm City*

But oh, I'm just a soul whose intentions are good

Oh Lord, please don't let me be misunderstood

—**Nina Simone, "Don't Let Me Be Misunderstood"**

Baltimore, Maryland is the 31st largest city in the United States. The city has a long history as an important seaport and is home to Fort McHenry, which sits at the mouth of Baltimore's Inner Harbor, the birthplace of the U.S. national anthem, "The Star-Spangled Banner."

Baltimore's nickname, "Charm City," traces its origins back to 1975 from a creative collaboration of four of the city's leading advertising executives and creative directors: Dan Loden and art director Stan Paulus of VanSant/Dugdale and Herb Fried and writer Bill Evans from W. B. Donor. Then-Mayor William Donald Schaefer asked the quartet to "Come up with something to promote the city. And do it now! I'm worried about this city's poor image" [7].

The nickname came from a line the four penned, "Baltimore has more history and unspoiled charm tucked away in quiet corners than most American cities out in the spotlight." Shortly thereafter, the name "Charm City" was born [7].

Most people either know Baltimore for its seafood, its two major sports teams (MLB's Baltimore Orioles and NFL's Baltimore Ravens), or the critically acclaimed show "The Wire," a drama based on the experiences of a former Baltimore homicide detective and public school teacher. With scores of delicious crab houses, a myriad of things to do on the water, and plenty of entertainment, the city has no problem letting those things represent them. However, it is the fictional drama that has taken a hold in pop culture, and the American conscience which has saddled the municipality with a less than deserving reputation.

At 21.8%, Baltimore's poverty rate is nearly twice the national poverty rate of 11.8%. The median household income is $48,840, which is significantly lower than the national median of $61,937 and that of neighboring Washington, D.C. ($92,266) [8]. Baltimore also has the lowest rate of mobility out of poverty in the country and a 20-year gap in life expectancy between its richest and poorest neighborhoods [8] (p. 8). However, this less than optimistic outcome is not shared universally by the citizens of Baltimore. Talk to them and they will acknowledge the bad rap the city has but quickly point to the city's being a place that has not discovered its charm. Ask them and they will tell you it is a city ripe for revitalization.

### 3.3. The Revival Story

**Impact** im·pact *noun*

1. the action of one object coming forcibly into contact with another.
2. the effect or influence of one person, thing, or action, on another.

Hotel Revival is a 107-room, 14-story boutique hotel nestled in the heart of Baltimore's historic Mount Vernon neighborhood. The building was once a private mansion owned by an art enthusiast, and formerly the site of the Baltimore Museum of Art's inaugural exhibition in 1923 [9]. It is fitting that Revival is Baltimore's only boutique art hotel. The term "revival" means renewal, rebirth, and restoration, and the hotel deems their purpose as a direct reflection of their name.

Spearheading the Charm Offensive are Donte Johnson, General Manager, and Jason Bass, Director of Culture and Impact (Figure 1).

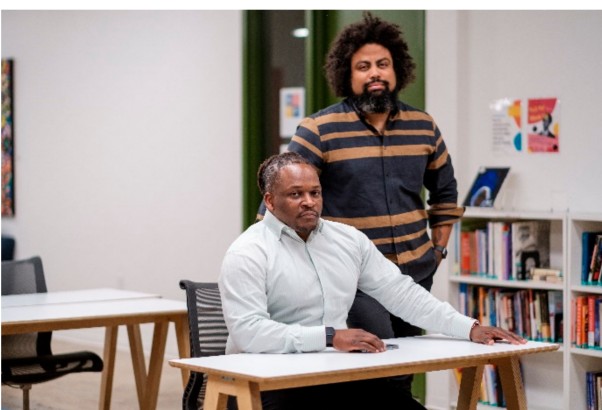

**Figure 1.** Donte Johnson & Jason Bass.

Donte Johnson brings more than twenty years of experience in the hospitality industry to Revival. Johnson is from Washington, D.C., a city that, while forty miles away, casts a large shadow over Baltimore. His partner in impact, Jason Bass, was born in Baltimore, but spent most of his childhood growing up in California, before moving back. As such, one of the ties that binds the dynamic duo is their being transplants. The two make it abundantly clear that they have embraced the city, and their goal is to help the city embrace itself.

The pre-pandemic vision the pair had for the hotel and surrounding community was "Creation." The pair saw it as an opportunity to help uncover Baltimore and create hand-shaking opportunities across socio-economic and geographic boundaries. Then, COVID-19 happened, which forced them to reimagine their concept (Figure 2).

Traditional hotels are focused on "Heads in Beds" and "Food & Beverage", which on average accounts for approximately 70% of a hotel's revenue. Since people were not partying, drinking, and eating, the pandemic created an opening for a new form of hospitality, one that was not just about the bottom line, but about purpose and people. To serve this double bottom line, they would have to devise a new way to deliver hospitality to guests, one that included the community and its people.

Social innovation is "a novel solution to a social problem that is more effective, efficient, sustainable, or just than present solutions and for which the value created accrues primarily to society as a whole rather than private individuals." [10] Innovation can involve creating new products, instituting new programs, or enacting policy changes. It requires generating, testing, and adapting these novel solutions.

> "We have great results as a business perspective and do things in a way that leaves the world in a little bit of a better place than we found it. We believe part of our purpose and our mission is to put smiles on peoples' faces." —**Donte and Jason**

The first group of people they used to "test" their theory were service industry workers and the community.

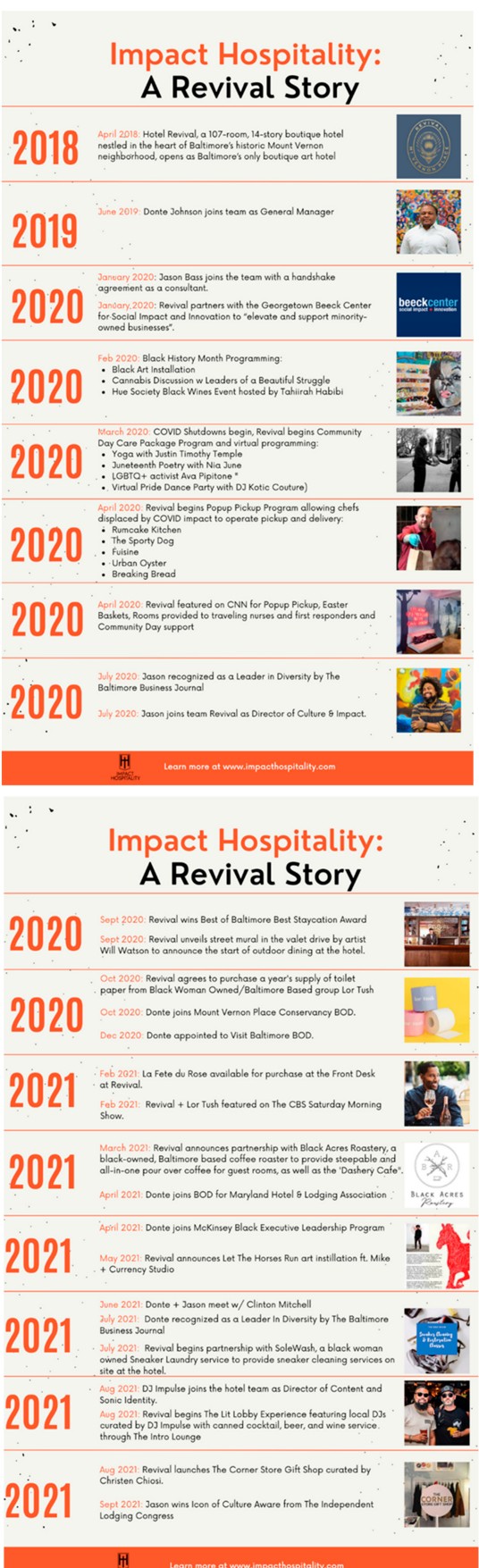

**Figure 2.** Hotel Revival Timeline.

### 3.4. Impact Hospitality: Changing the Face of Hospitality

**Social Impact** so·cial im·pact *noun*

Social impact is a significant, positive change that addresses a pressing social challenge [11].

The underlying goal of Donte and Jason's work is to create social impact by leveraging their networks, expertise, and resources. Creating social impact is the result of a deliberate set of activities with a goal to improve the conditions and current situation through structural movement of the status quo. At Revival, that means centering the mission as their purpose and demonstrating their commitment to the ongoing restoration of Baltimore. The hotel states proudly on their website that,

> "Our tight knit community is what makes Baltimore 'Charm City.' To support the people and businesses around us, we create day-to-day programs where local businesses are able to use our space to raise money for nonprofits in the area. We make it a point to introduce our guests to Baltimore by building spaces with services and products from our community" [9]

In March 2020, when shutdowns due to the rapid spread of COVID-19 began, Hotel Revival decided to soldier on with their mission. They responded with several initiatives, including the Community Day Care Package Program, offering supplies, bagged lunches, and fresh produce from Coastal Sunbelt Produce and Hungry Harvest to individuals in need. They also provided free rooms for military personnel, police officers, firefighters, and traveling nurses, all of whom needed rest as frontline workers in the fight against the coronavirus. The hotel also instituted virtual programming, such as yoga with Justin Timothy Temple, Juneteenth Poetry with Nia June, and a Virtual Pride Dance Party with DJ Kotic Couture. Revival was determined to stand tall as a beacon in the community during some of the darkest days of the pandemic.

Baltimore is faced with impediments and challenges that obscure its beauty and create barriers to success that are difficult to overcome, which was exacerbated by a once in-a-generation global crisis. Hospitality and tourism have long been opportunities for guests and residents of a city to serve as ambassadors to the world.

> "Travel exposes you to new languages, traditions, history and provides an opportunity to gain a new perspective from the locals and community of a specific place and culture. The more people who travel, the more understanding and compassionate of a world we live in." —**Visit Baltimore President/CEO, Al Hutchinson**

However, with travel being greatly reduced, there are very few envoys to foster diplomatic relations between the city, its residents, and the rest of the world, thereby creating a pressing social challenge. To meet this challenge, Donte and Jason devised a plan:

- **Create**: Create space for the intersection of thoughts, ideas, and peoples in an imaginative collaborative environment.
- **Develop**: Organically develop an ecosystem for connections and partnerships.
- **Expose**: Leverage space and opportunities to expose Baltimore to itself through cultural exchanges that would not normally and naturally occur.
- **Strengthen**: Fortify the community by improving access to assistance, addressing their needs, and elevating their voices.

It is through the implementation of this plan that a new form of hospitality was formed.

### 3.5. Impact Hospitality

**Impact Hospitality** im·pact hos·pi·tal·i·ty *noun*

A significant, positive influence on guests, visitors, and the community created as the result of a deliberate set of activities with the goal of addressing a pressing social challenge. This can be achieved through the creation of space for conversation, connections, and cultural exchanges.

As entrepreneurs and two individuals in love with the city of Baltimore, it was clear to Donte and Jason that one of the best ways to connect the city was to enlist the support of its entrepreneurs and small businesses. Baltimore's entrepreneurship scene has experienced a rapid rise over the past decade and was recently ranked by INC as the 38th best city to start a business.

According to the 2020 Annual Business Survey (ABS), covering reference year 2019, approximately 18.7% (1.1 million) of all U.S. businesses were minority-owned. Of that number, approximately 12% of minority-owned businesses were Black-owned [12]. Black business owners have been disproportionately affected by the economic downturn spurred by the pandemic, partly because they were more likely to already be in a precarious position, including being more likely to be in communities with business environments that are more likely to produce poor business outcomes.

Only four percent (4%) of Black-owned businesses survive the start-up stage, even though 20% of Black Americans start businesses [13]. Even if they survive the start-up stage, Black-owned businesses still disproportionately struggle with debt and raising capital in addition to challenges such as a lack of helpful relationships in the business community. Start-up capital is associated with better business performance, but Black entrepreneurs have less of it. Black entrepreneurs start their businesses with about $35,000 of capital, white entrepreneurs $107,000 [13] (p. 10). Such obstacles make Black-owned businesses less likely to survive and grow.

The Federal Reserve Bank of New York published a report that found that about 58% of Black-owned businesses were at risk of financial distress before the pandemic, compared with about 27% of white-owned businesses [14]. Additionally, the report found that the number of active business owners fell by 22% from February to April 2020, the largest drop on record. However, the decline is even more striking when viewed across racial and ethnic groups. For example, Black businesses experienced the most acute decline with a 41% closure rate, Latino business owners fell by 32%, and Asian business owners fell by 26%. In comparison, the number of white business owners only fell by 17% [14] (p. 1). In a city that is 62% Black, this is particularly profound and potentially debilitating.

Like most of the country, the residents and companies of Maryland and Baltimore have experienced significant economic difficulties as a result of the pandemic. In the beginning of 2021, nearly 500,000 Marylanders had filed unemployment claims. This represents more than 15% of Maryland's labor force [8] (p. 22). Major Baltimore area employers have laid off thousands of employees. Such companies range from Under Armour to Horseshoe Baltimore Casino to Bloomin' Brands.

> "Black and brown communities have long endured inequities from policies rooted in bigotry and structural racism—and COVID-19 has only exacerbated those known disparities. We must have intentionality in addressing our challenges with an all hands on deck approach. Partnership between public, private and nonprofit sectors is critical. Through the leadership of Donte and Jason, we are harnessing the power of partnerships and community empowerment here in Baltimore. I commend them for their work to drive social impact and supporting and empowering Black and brown entrepreneurs." —**Nick Mosby, President of the Baltimore City Council**

*3.6. Measuring Impact Hospitality*

One of the most challenging aspects of social impact is measurement. This is in part because it includes assessing complex concepts while accounting for external factors that might have influenced the element you are trying to measure, and it requires long-term consistency operating with a robust set of tools within a comprehensive framework to accurately evaluate.

> "[C]alculating the "good" done is tough. First because knowing what "good" means is hard, secondly because relating "good" to dollars is like translating a symphony into organic chemistry, and third because identifying cause and effect

is tough (did your grant create more jobs, or did the economy just happen to get better?)." —**Sean Stannard-Stockton, "Social Return on Investment," Tactical Philanthropy** [15]

Measuring social impact affords one the opportunity to acquire quantitative and qualitative inputs and provides a baseline measurement for an organization's activities. Sir Ronald Cohen wrote, "Calculating the true costs of social problems, then measuring the costs of interventions against them means organizations could quantify their impact in financial terms" [16]. Furthermore, measuring provides accountability to stakeholders, be it hotel ownership, guests, or the community at-large.

Donte and Jason were intentional about which brands and companies the hotel partnered with for the procurement of supplies such as toilet paper, coffee, and wine. To evaluate their progress and impact they utilized an Impact Tracker designed by Georgetown University's Beeck Center for Social Impact and Innovation to assess their procurement spending, family stability, talent acquisition, sales, and earned media. The goal was to measure every possible datapoint by which the team was effectuating positive social change. (Figure 3)

| Demographic | Jan |
|---|---|
| # of Black owned | |
| # of Women owned | |
| # of LGBTQ+ owned | |
| **Location** | |
| # of vendors within city limits | |
| # of vendors within low-income/HUD areas | |
| # of vendors within state | |
| **Programming** | |
| # of activities/events aligned with impact focus areas | |
| # of activities/events supporting minority interests | |
| **Sales** | |
| # of customers influenced by impact efforts | |
| # of groups influenced by impact efforts | |
| **Internal** | |
| # of black people in leadership roles | |
| # of women in leadership roles | |
| # of employees living in low-income areas | |
| # of employees living in city limits | |
| % of employees above minimum wage | |
| **External** | |
| # of referrals of minority suppliers/vendors to other organizations | |
| **Media** | |
| # of related local media hits | |
| # of related global media hits | |

**Figure 3.** Georgetown Beeck Center Social Impact Tracker.

Admittedly, sales are difficult to measure during a pandemic as the hospitality and tourism industry is suffering and will be for quite some time. Yet, market share at Hotel Revival for September 2021 was the highest in the short history of the hotel, which opened May 2018. This number eclipsed the hotel's previous record set in August 2021 and is up 77% over the baseline year (September 2019). "Beyond the total commercial performance

of the asset, business generated as a direct result of our culture & impact programming is well into six figures and growing," Donte Johnson.

Labor issues created and aggravated by COVID-19 are among the greatest threats to the U.S. economy. Because many hospitality employees are from marginalized communities, which have borne the brunt of COVID-19 in America, many jobs have been lost and many workers have sought employment in other industries. The labor shortage in hospitality is atop the list of concerns of every level of the industry from CEOs to the American Hotel and Lodging Association (AH&LA), to property leaders, across markets and segments.

However, Revival is not experiencing the difficulty the rest of the country and industry is suffering in attracting and retaining talent. While the industry is starving for talent, their time to fill open leadership positions is at a historic low (21 days). As of this writing, there are no data points to explain this phenomenon. However, Donte and Jason believe their good fortune is a testament to their pioneering model. They believe that people want to work for a company that does not just portray itself as good but is actually doing good work in and around the community.

One aspect of measurement the pair have emphasized is storytelling, which can be seen on the tracker as Earned Media. Earned media refers to publicity gained through promotional efforts other-than-paid media advertising. Donte and Jason believe storytelling is acutely important to drive awareness. "What it means to be a storyteller on behalf of a community is more than just what we say to people in the lobby, but what we also say to people in their experience." Given the immense amount of work they have done in the community, there are many stories to be told from local candle makers to creative oyster cuisine, to plant styling and care. In 18 months, the hotel has tallied more than $12 million in earned media. From January 2020 to the ensuing 18 months, the brand had an astronomical 1.362 billion impressions related to social impact, making it clear that their message is being spread far and wide and is resonating.

Small businesses are the lifelines of communities, so empowering them naturally empowers the people within the communities they serve. One way that brands in hospitality and tourism can do so is through inclusive procurement practices. Dedicating funds to minority-owned and women-owned businesses can help minority-owned small businesses. It is offering a hand up to qualified individuals who, without the help, would likely never have the opportunity to demonstrate their ability.

### 3.7. Community Stories

3.7.1. Your Tush Deserves the Best: Lor Tush

Lor Tush is an organic bamboo toilet paper company founded by the two youngest of 4 children of Nigerian immigrants, Nnadagi Isa and Louise Isa (Figure 4). They got their start in entrepreneurship prior to the pandemic and were able to flourish during it. When asked, "Why toilet paper?" Co-Founder Nnadagi Isa stated that the business was born out of necessity: The need to find an organic sustainable option for your delicate parts that is conducive to their environmental and food allergies. They also stated that the brand was created because of their desire to create something of their own, and toilet paper was something they thought could achieve those ends.

They formed the idea in 2018 and developed their first batch of rolls in November 2019, just months before COVID-19 would be declared a global crisis. The result was a toilet paper that was "free from dyes and other chemicals and was also safe for septic tanks," and that also, importantly, felt good. For the sisters, who born in DC but moved to West Baltimore for their dad's job in high school, the enterprise is also a bet on Baltimore. Nnadagi said, "Out of all the places we moved to, Baltimore made me feel like I belonged. The DIY scene, Creative Culture, made [us] feel like home."

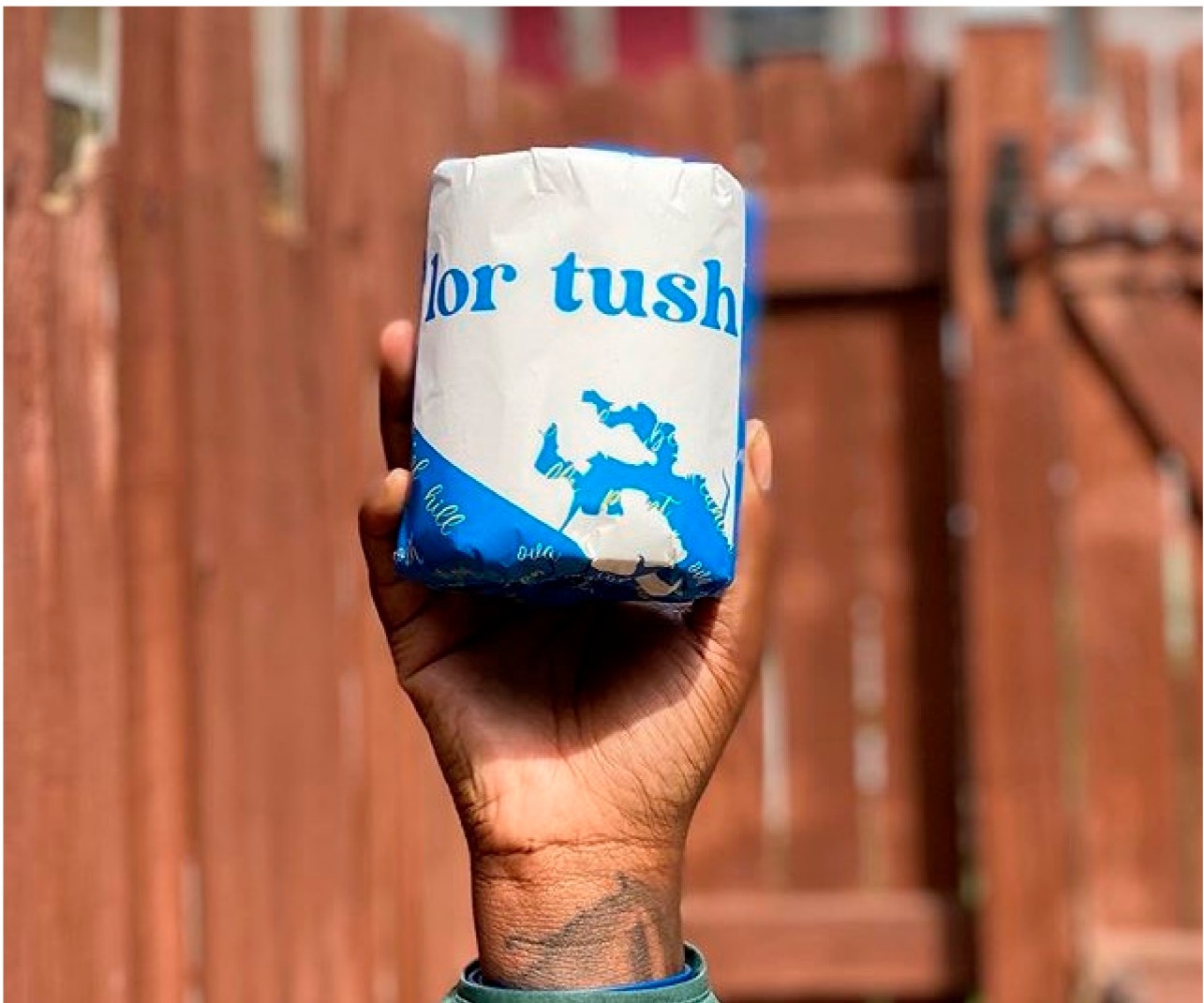

**Figure 4.** Lor Tush.

Running and operating a fledgling small business is daunting at any time and is perilous during a pandemic. However, the pair did not back down and accepted the challenge. At the outset of the pandemic, toilet paper was hard to come by due to a run on it and other hygiene products, leaving many without, particularly vulnerable populations. During the first run of rolls, they donated 5000 rolls of toilet paper in the city that made them feel liked they belonged. To date, they have donated 15,000 rolls of toilet paper to community fridges and pantries in the area.

While the idea and execution has been all Nnadagi and Louise, they credit the help of Donte and Jason for putting them on a trajectory that they admit they were not sure they could reach. Nnadagi and Jason met via a mutual friend after a conversation about e-bikes. Their mutual friend shared with her that Jason, whom she had not met yet, was looking for an e-bike, and as the owner of an e-bike, she offered to be of assistance. Her first text to Jason was, "Do you want to start a bike gang?" What followed was not a rogue posse of leather jacket-wearing e-bike riders, but a conversation that revealed Nnadgi and her sister started a company and that he should try their toilet paper. After sampling the goods, himself, Bass invited the pair to the hotel for a meeting with Donte and the Hospitality Manager. After working on the finer points of the agreement, Lor Tush soon became the official toilet paper supplier of Hotel Revival. If that was not enough, the subsequent press the hotel facilitated helped the brand acquire nationwide acclaim, which led to their briefly selling out after a guest spot on "CBS This Morning."

When asked about why the hotel chose to go with Lor Tush versus a bigger, more well-known brand, Donte replied, "When we're out in the neighborhood handing out groceries to the community, the owner or C.E.O. of your major toilet paper provider isn't elbow-to-elbow with you, but they were. If given the opportunity, I'm going to go with local small businesses because of the opportunity to multiply the impact."

### 3.7.2. The Best Part of Waking Up: Black Acres Roastery

Black Acres Roastery Founder, Travis Bell, noticed something missing from his Baltimore neighborhood: coffee. Bell stated, "We didn't have coffee in the neighborhood, and I've always gravitated towards food, and I wanted to do something outside of my day job (Occupational Therapist)." Bell headed to Minnesota to take roasting courses and fell in love with the science, craftsmanship, and creativity of coffee making. He started the business roasting out of a wellness space by doing events with the coffee. In 2019, he started doing online sales and wholesale, and the company grew (Figure 5).

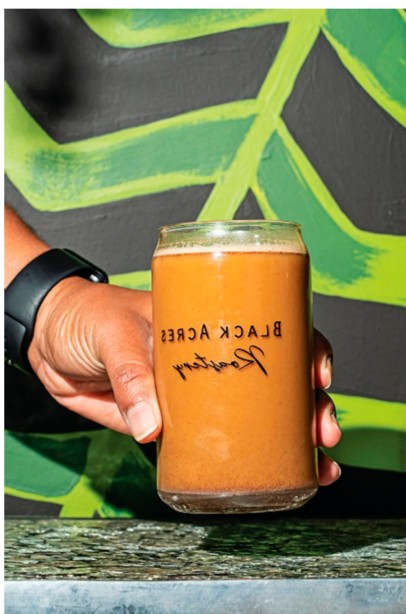

**Figure 5.** Black Acres Roastery.

Bell believes that Baltimore gets left behind in many conversations, particularly those about Black entrepreneurship, and he wants to be a part of the conversation. He saw coffee as an opportunity because of its inherent diversity in taste and form and because he did not think Baltimore was being well-represented in that space, which provided an opening for Black Acres Roastery. "You can't sit on your hands; If you want something you'll make the effort to put the time in," said Bell.

Travis connected with Hotel Revival through his relationship with Jason Bass, whom he knew through Bass' other business ventures. Travis, confident in his product, knew where he needed help the most: getting inside the room where it happens, because once inside, the coffee would handle the rest. Once GM Donte Johnson tasted the coffee, they crafted a partnership. Then, COVID-19 happened. What followed was a year of uncertainty, trepidation, and silence. When the time was optimal to reengage, Jason and Donte kept the promise they made to Travis. Today, the hotel's rooms and suites feature Black Acres Roastery single-serve steepable coffee bags which can be steeped in hot or cold water, are 100% biodegradable, and the outer packaging is made using renewable and compostable materials. The company also provides coffee for the hotel's coffee shop, "The Dashery".

"Their desire and passion to help others explodes through the conversations, particularly allowing Black makers to shine. Them being able to put people on a platform to do what they're meant to do has been exceptional. Not sure if anyone

else could have done what they did especially during these unprecedented times, putting others before themselves to put a community up higher than it was the day before." —**Travis Bell, Founder Black Acres Roastery**

3.7.3. A Rosé by Any Other Name Wouldn't Taste as Good: La Fête du Rosé

With degrees in Mathematics (B.S.) and Industrial Engineering (M.S.), Baltimore (Randallstown) native Donae Burston, is probably the last person you would expect at the helm of one of the fastest growing rosé brands in the country (Figure 6). But as the founder of La Fête du Rosé, that is exactly who he is. So, how does one make the leap from information technology to wine entrepreneur?

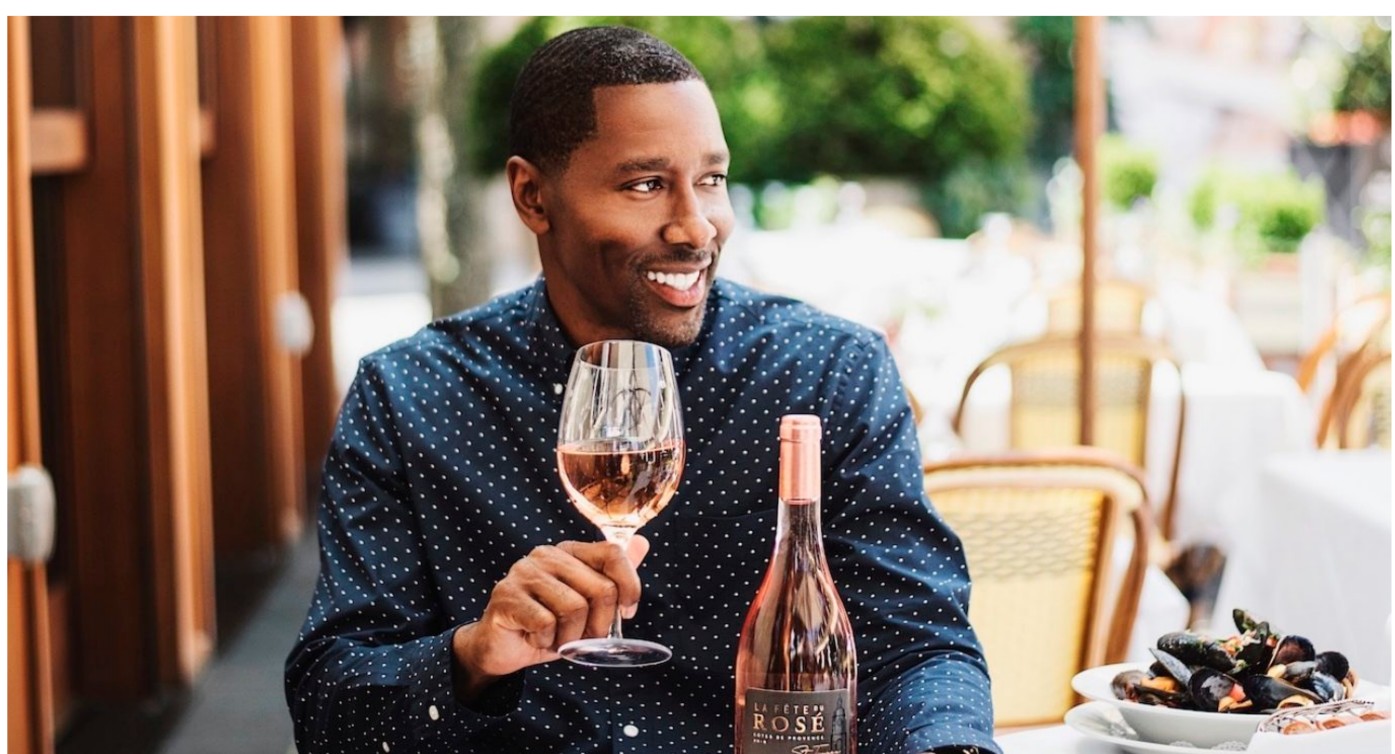

**Figure 6.** La Fête du Rosé.

To celebrate his 30th birthday, Burston and a few friends visited St. Tropez, France. While there, he saw lots of pink bottles being poured, which at the time he mistook for white zinfandel. After getting a taste for himself, he quickly realized two things: 1. that he had never had anything like it; and 2. it was not white zinfandel at all. It was rosé!

Years later, when the rosé boom hit the United States, Burston noticed something strange. He saw that the delicious beverage he came to know and love was only be marketed to white women, which was counter to his experience in France, where everyone drank it. So, he sought to do something different and share his love of rosé with the world. In 2019, Burston launched the first entirely Black-owned rosé, out of St. Tropez, France, La Fête du Rosé. He estimated that the brand would sell approximately 50,000 cases of rosé in 2021.

Hotel Revival was the first on-premise hotel for La Fête du Rosé, and it started with just a taste. From there, Donte and Jason introduced the brand to the national team in the Hyatt group and helped tell the story of the brand, throughout the city and beyond.

An integral part of the brand's story is their philanthropy. As a proud product of Randallstown, one of Burston's central issues has been the lack of positive storytelling by corporate America about the place he calls home. Burston believes that there are many paths to being successful and that for many of the city's residents, all it takes is changing the idea of what is possible. Being the man of action that he is, Burston and the company

donate to Learning to Live Movement, Inc, an organization that, since 2015, has been providing travel opportunities for high school students of color. Five dollars of every bottle purchased at Hotel Revival goes towards this organization, which is separate from the personal and corporate donations Donae and the company make.

One of the things Burston appreciates about his partnership with the hotel is how forthcoming and charitable Donte and Jason have been with leveraging their network to help others. He believes that Baltimore is changing and growing and that he and others throughout the city who look like him should take more ownership of that success.

3.7.4. The Artist Currently Known as Will Watson

Will Watson has wanted to be an artist since he was 5 years old. As a native of Indianapolis, he knew that to pursue his dreams, he would have to leave the comforts of home, so he moved to Washington, D.C. to pursue his passion. When things did not work out, he knew he did not want to go back home, so, at the urging of some friends, he decided to go to graduate school at Maryland Institute College of Art (MICA), where he graduated with a Master's in Fine Arts.

The two years post-graduation did not quite pan out the way Watson wanted, which found him in 2020 at a crossroads. In the wake of everything going on with COVID-19 and social unrest over the police-involved murders of Breonna Taylor and George Floyd, he needed to get out. He needed to create. In the Summer of 2020, this impetus led him to Graffiti Alley, where he spraypainted a portrait of Breonna Taylor which caught the eyes of Jason and Donte. (Figure 7)

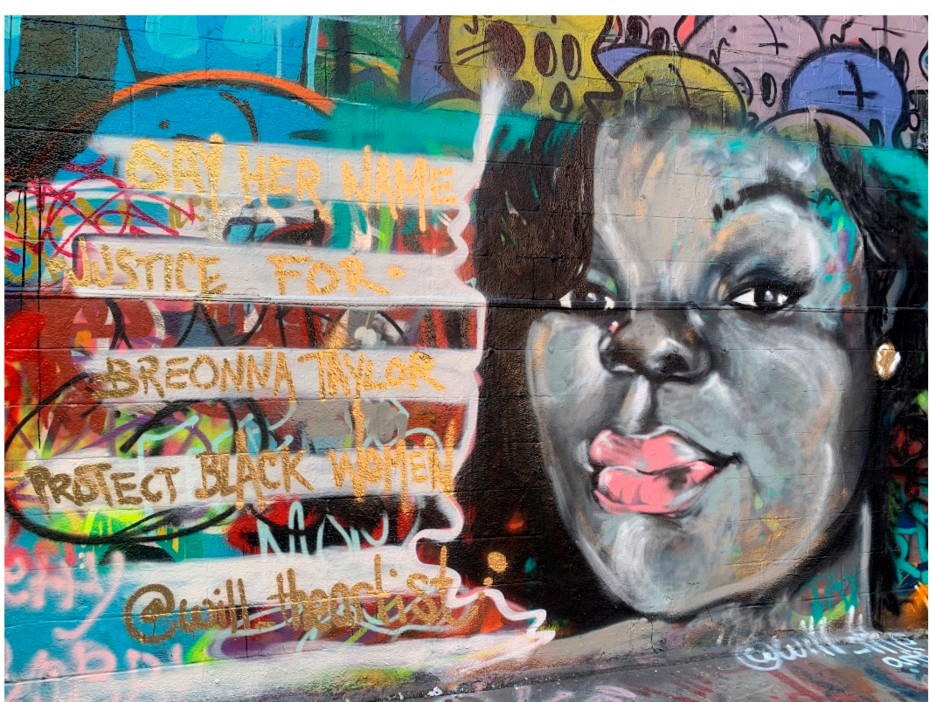

**Figure 7.** Breonna Taylor Mural by Will Watson.

Will went from just putting his feelings on "canvas" to having a viral moment in the city. It was overcoming his fear to focus on sharing his gift that led him to the alley that fateful day in 2020, which subsequently resulted in a commissioned piece on behalf of Hotel Revival. The project was a street mural in the valet drive to announce the start of outdoor dining at the hotel. Watson says, "That project kept me in Baltimore. Because between COVID and my financial situation, I was unsure if I could stay." While fear and finances may have initially kept him in Baltimore, the opportunity to share his talent kept him here.

As an artist, Will is a very solitary person because he spends an inordinate amount of time creating. He credits Donte and Jason for getting him out of the house. He notes that a current collector of his work saw one of his paintings at the hotel and contacted Will ready to buy it.

> "As someone who's not from this place, to meet people from a place that you're not from and for them to take you in and treat you like family, or someone that they've known forever has been great. They're good people to have on my side and I thank God that I've been able to meet those type of people. I prayed for good people along my journey and that's what I've been getting." —**Will Watson, Artist**

There is a gap between the haves and have nots, and in places like Baltimore, where the unemployment rate exceeds the state and national average, that gap can feel like an ocean. Workers of color, who are unemployed at three times the rate of white workers, may feel like they are lost at sea. However, Maryland ranked first of any state nationally for per capita business ownership among women and people of color [8]. Add that to Hotel Revival's partnerships with small and minority owned small businesses and entrepreneurship may be seen as a lifeline.

> "Many of the things we do have no tangible impact on the financial bottom line of the business, but we do them because we're uniquely positioned to do them. Businesses can move in communities differently and we're taking the opportunity to lead the charge and demonstrate." —**Donte Johnson, GM**

## 4. Conclusions

Donte Johnson and Jason Bass are intelligent and talented individuals with infectious personalities. They lead with compassion and care, centering Baltimore in everything they do. Impact Hospitality was not born from a think tank looking for the next trend in the industry. It was birthed from love and fear—love of a city that has welcomed them, and fear that due to circumstances beyond its control, its charm would never be truly revealed. They live by the idea that, "When you invest in the city and its people, it demonstrates a level of connection to the community." That connection is evident in every partnership, collaboration, guest stay, and space rental.

As the country continues to grapple with COVID-19 and its resulting issues (supply chain disruption, labor shortage, new variants, and vaccine hesitancy), it is clear that this virus will be with us for some time and, as such, we will need to adjust to a new normal. Now is an excellent time to innovate and create sustainable practices in the hospitality industry of tomorrow through Impact Hospitality.

The duo is in lockstep with their vision of Impact Hospitality and their belief that this approach can be replicated beyond the walls of Hotel Revival. However, one thing they are not in sync about is a constant refrain Donte has echoed during their numerous media interviews: "Everyone can be a Revival." To the contrary, Jason does not believe everyone can be Revival, nor should they endeavor to be Revival. Jason adds, "If every hotel could, and truly wanted to, they would. Instead, what we hope we have shown the world is the prototype from which to borrow and draw inspiration, so that others can emulate the impact in their own communities."

Shortly before publication, the pair shared evidence that the blueprint is working in the form of the hotel's 2021 Annual Report detailing the media and public relations coverage. Revival had a total ad value estimate of over $25 million in 2021, which was a 495% change from the previous year. The brand received over 2 billion hits, with the majority of coverage (57.4%) being in the area of impact, which is not only phenomenal but also a testament to the need for and success of impact hospitality.

Donte and Jason believe that to effectuate significant, positive influence on guests, visitors, and the community, the plan to Create, Develop, Expose, and Strengthen must be paired with your city's distinctive DNA, an unwavering focus on profit and purpose, and a commitment to creating social impact to produce Impact Hospitality.

**5. Trademarks**

Impact Hospitality™.

**Funding:** This research received no external funding.

**Institutional Review Board Statement:** Not applicable.

**Informed Consent Statement:** Informed consent was obtained from all subjects involved in the study.

**Acknowledgments:** I would like to acknowledge Donte Johnson, Jason Bass, Lor Tush, Will Watson, Black Acres Roastery, La Fête du Rosé.

**Conflicts of Interest:** The author declares no conflict of interest.

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
