# Peer review of "Impact Hospitality: Creating Social Impact through Hospitality"

_sustainability, doi:10.3390/su14106274_

Round 1
Reviewer 1 Report
The objective of the paper: This paper seeks to define a new form of hospitality designed to serve the double bottom line of profit and people. This body of work chronicles the stories, lessons learned and the path ahead for an industry in need of a way forward to create: Impact Hospitality.
The authors have demonstrated that they can measure Impact Hospitality using Donte and Jason plan (CEDS) through "Impact Tracker":
• Create: Create space for the intersection of thoughts, ideas, and people in an imaginative collaborative environment.
• Develop: Organically develop an ecosystem for connections and partnerships.
• Expose: Leverage space and opportunities to expose Baltimore to itself through cultural exchanges that would not normally and naturally occur.
• Strengthen: Fortify the community by improving access to assistance, addressing their needs, and elevating their voices.
Line 28: The specific statement (more than 80 countries) and claim (involving 1.5 billion children) should be backed by a reference.
Comment: A general statement maybe use (e.g. majority of countries; involving school-going children).
Line 97 - 123: Five paragraphs from the same source (American Hotel & Lodging Association. (2021, July). State of the Hotel Industry 2021. Washington, D.C.: American Hotel & 547 Lodging Association.).
Comment: Suggest using or adding a table related to the above information and citing it from one source.
Line 119: As the world attempts to recover, hotels were expected to add 200,00 direct hotel
Comment: (200,00 or 200,000?)
Line 124: Baltimore: Charm City provided a brief and interesting background of the case however there is a need to relate the impact of the pandemic on Baltimore in general.
Suggestion: Add a paragraph that provides a story / a picture of the actual situation faced by Baltimore during the time the case was written. This may further enhance understanding and the importance of Impact Hospitality
Line: 161 Figure A1 hard to read fonts/size
Comment: Redo Figure A1 (hard to read)
Line 186: Purpose and people are the double bottom line to explain the concept of Impact Hospitality, and both meanings are important to be captured from Donte and Jason's work.
Comment: Elaborate "purpose" and "people" after line 189.
Line 236: Impact Hospitality im·pact hos·pi·tal·i·ty noun. A significant, positive influence on guests, visitors, and the community was created as the result of a deliberate set of activities with the goal of addressing a pressing social challenge. This can be achieved through the creation of space for conversation, connections, and cultural exchanges.
Comment: One important element pointed by the authors for this innovative idea to work is selection of partners (refer to line 229; 280; 282; 412; 455). Suggest to add the word "partners" or "partnership" in the "Impact Hospitality" working definition. It was also highlighted in Line 304.
Line 286: Measuring Impact Hospitality (social impact measurement). Line 304 brands and partner companies were measured using "Impact Tracker" Line 380 The goal was to measure every possible data point by which the team was effectuating positive social change.
Comment: Can the process be repeated? Example of data point or "the "Impact Tracker" report to understand the measuring concept. Example Line 332 storytelling (Earned media), is there another data point? The highest hits will be reported as "Community Stories".
Line 353: Lor Tush. Reported well
Line 395 Black Acres Roastery. Reported well
Line 430: La Fête du Rosé. Good
Line 451: Learning to Live Movement, Inc
Line 459: Will Watson
Comment: At the end, a paragraph to connect all the above stories to Hotel Revival and provided inputs to measuring the concept of "Impact Hospitality"
Reviewer 2 Report
This is a case study written in the 1st person narrative. It is not a research study in the classical sense. That distinction should be clearly made in the title, abstract, and introduction. Yes, case studies are useful. Yes, rich narratives regarding one situation can be enlightening. However, the framework presented initially doesn't clearly make that intention known to the reader. There are subjective statements by the author which come across as opinions rather than informational. The case study should be about the two leaders and their unique approaches, not what the author judges regarding the case. Allow the readers to reach their own conclusions. In summary, my recommendations for improvement are (1) reframe the title, abstract, and introduction to make clear that this is a case study as well as to more succinctly present the key findings; and (2) remove the author's judgments about the activities of the two leaders so that readers can reach their own assessments. By presenting results in an unbiased manner, readers can do that.
Reviewer 3 Report
Thank you for the opportunity to revise the manuscript titled "Impact Hospitality: Creating Social Impact Through Hospitality" that was submitted to Sustainability. The relevance of the research realized by the author of the paper is obvious. This is a very informative case report on doing business in the context of the COVID-19 pandemic. It was interesting to read an inspiring story about minority communities and local small businesses developed in difficult circumstances. The ideas, effort and energy invested have resulted in different sustainable practices in the hospitality industry.
However, the main and significant drawback of this article is section 3 - Results. It is not common for an entire section to consist of a list of figures. Please try to incorporate some text with the main results of your research.
I suggest publishing the paper as a case report after corrections.
Round 2
Reviewer 2 Report
Repositioning made all the difference.
This manuscript is a resubmission of an earlier submission. The following is a list of the peer review reports and author responses from that submission.